# Sticking to the Rules: Outcome and Success Rate of Guideline-Based Diarrhea Management in Metastatic Breast Cancer Patients Treated with Abemaciclib

**DOI:** 10.3390/jcm12051775

**Published:** 2023-02-23

**Authors:** Flavia Jacobs, Elisa Agostinetto, Alessandra Solferino, Rosalba Torrisi, Giovanna Masci, Armando Santoro, Rita De Sanctis

**Affiliations:** 1Medical Oncology and Hematology Unit, IRCCS Humanitas Research Hospital, Humanitas Cancer Center, Via Manzoni 56, 20089 Rozzano, MI, Italy; 2Department of Biomedical Sciences, Humanitas University, Via Rita Levi Montalcini 4, 20090 Pieve Emanuele, MI, Italy; 3Academic Trials Promoting Team, Institut Jules Bordet, L’Université Libre de Bruxelles (U.L.B.), 1070 Bruxelles, Belgium

**Keywords:** diarrhea, supportive care, breast cancer, CDK4/6 inhibitors, abemaciclib

## Abstract

In clinical trials testing abemaciclib in patients with hormone-receptor-positive (HR+), HER2-negative (HER2-) advanced breast cancer, diarrhea is a very common adverse event (occurring in approximately 85% of patients, any grade). Nonetheless, this toxicity leads to abemaciclib discontinuation in a small proportion of patients (approximately 2%) thanks to the use of effective loperamide-based supportive therapy. We aimed to determine whether the incidence of abemaciclib-induced diarrhea in real-world trials was higher than the one reported in clinical trials, where patients are highly selected, and to evaluate the success rate of standard supportive care in this setting. We conducted a retrospective, observational, monocentric study including 39 consecutive patients with HR+/HER2- advanced breast cancer treated with abemaciclib and endocrine therapy at our institution from July 2019 to May 2021. Overall, diarrhea of any grade occurred in 36 patients (92%), of whom 6 (17%) had diarrhea of grade ≥3. In 30 patients (77%), diarrhea was associated with other adverse events, including fatigue (33%), neutropenia (33%), emesis (28%), abdominal pain (20%), and hepatotoxicity (13%). Loperamide-based supportive therapy was administered to 26 patients (72%). Abemaciclib dose was reduced in 12 patients (31%) due to diarrhea, and treatment was permanently discontinued in 4 patients (10%). In 58% of patients (15/26), diarrhea was effectively managed with supportive care and did not require abemaciclib dose reduction and/or discontinuation. In our real-world analysis, we observed a higher incidence of diarrhea related to abemaciclib compared to data from clinical trials, and a higher rate of permanent treatment discontinuation due to gastrointestinal toxicity. Better implementation of guideline-based supportive care could help to manage this toxicity.

## 1. Introduction

Breast cancer is the most commonly diagnosed cancer in women worldwide. Up to 2.3 million patients are diagnosed with breast cancer every year [1]. Overall, approximately 6–12% of breast cancer patients are metastatic at the time of diagnosis, and nearly one-third of patients diagnosed at an early stage develop metastases during their lifetime [2]. Hormone-receptor-positive (HR+)/HER2-negative (HER2-) is the most common breast cancer subtype and accounts for more than 70% of all new diagnoses [3].

Cyclin-dependent kinase (CDK)-4/6 inhibitors represent a major milestone in the treatment of HR+/HER2- advanced breast cancer and have revolutionized the current treatment landscape of this disease. Indeed, the addition of palbociclib, ribociclib, or abemaciclib to endocrine therapy in first or subsequent lines for patients with HR+/HER2- advanced breast cancer has demonstrated a clinically significant survival benefit, thus becoming a standard of care for this subset of patients [4,5,6,7,8,9]. Moreover, the addition of abemaciclib to endocrine therapy for the adjuvant treatment of high-risk HR+ patients has demonstrated a significant improvement in terms of invasive-disease-free survival (iDFS), leading to U.S. Food and Drug Administration (FDA) approval for this patient population [10]. Although CDK 4/6 inhibitors represent a well-tolerated and manageable class of drugs, they are associated with some potential adverse effects. Some of these, such as the hematologic toxicity (especially neutropenia), are common to all CDK4/6 inhibitors, although differences in frequency and severity exist. On the other hand, other adverse events are more characteristic of the individual CDK 4/6 inhibitors. For instance, ribociclib is more commonly associated with prolongation of the QTc interval and alteration of liver function, whereas gastrointestinal toxicity is more common with abemaciclib [11]. The absolute risk for any-grade diarrhea is far higher for abemaciclib than for palbocicib and ribociclib [12].

In a pooled analysis of the two phase III studies [8,13] with abemaciclib, diarrhea of any grade occurred in 84.6% of patients, and 11.7% of them experienced grade 3 diarrhea [14]. However, in most patients, diarrhea could be effectively managed with supportive medications (i.e., loperamide) or dose adjustments, ultimately resulting in a low rate of treatment discontinuation (less than 3%) without adversely affecting PFS outcome [14]. Although abemaciclib-induced diarrhea is usually mild to moderate, determining its frequency is critical as ineffective management of this adverse event is closely associated with poor patient adherence, reduced quality of life (QoL), and dose omission.

In clinical practice, the treatment of abemaciclib-induced diarrhea is well established: guidelines recommend educating patients about the potential occurrence of diarrhea and proactively administering supportive therapy at the first sign of loose stools. Dose interruption or reduction is recommended when diarrhea is grade 2 (persistent or recurrent) or higher [15].

In this study, we aimed to determine whether the incidence of abemaciclib-induced diarrhea was higher in real-world than in clinical trials, in which patients are rigorously selected. In addition, we investigated whether guideline-based supportive therapy was effective in treating abemaciclib-induced diarrhea in our real-world patient population.

## 2. Materials and Methods

### 2.1. Study Design

This is a retrospective, observational study of consecutive female patients with HR+/HER2- advanced breast cancer, treated with abemaciclib and endocrine therapy from July 2019 to May 2021 at Humanitas Research Hospital. The clinical charts of potentially eligible patients were retrospectively reviewed. Eligible patients were aged ≥18 years, with a histologic diagnosis of HR+/HER2- advanced breast cancer, treated for advanced disease in the first or later line of treatment with abemaciclib and endocrine therapy. Patients with short follow-up (i.e., <1 month) or unavailable safety data were excluded.

Baseline clinicopathological characteristics, information about treatment, outcomes, adverse events, and supportive care were collected from medical records.

The research proposal was approved by the local Ethical Committee (Independent Ethical Committee IRCCS Istituto Clinico Humanitas, protocol number ONC/OSS-03/2022). Written informed consent for treatment and use of clinical data for scientific purposes was provided by all patients. This study was conducted according to the principles of the Helsinki Declaration.

### 2.2. Study Population

All patients had a histologic diagnosis of HR+/HER2- advanced breast cancer and were treated with abemaciclib and endocrine therapy. HR status was assessed by immunohistochemistry (IHC). Estrogen and progesterone receptor (ER and PR) status was considered positive if expressed in >1% immune-reactive cells. HER2 status was assessed by IHC (0, 1+, 2+, or 3+ score) and by fluorescent in situ hybridization (FISH), the latter performed in all patients with a HER2 IHC score of 2+. HER2 overexpression positivity was defined according to ASCO-CAP guidelines [16] for a membrane staining IHC score of 3+ or 2+ with evidence of FISH amplification. HER2 IHC scores of 1+ and 0 defined a HER2-negative status. All patients received abemaciclib at the standard initial dose of 150 mg per os twice daily, administered continuously, in combination with endocrine therapy, either letrozole (2.5 mg per os once daily) or fulvestrant (500 mg intramuscularly on days 1, 14, 28, and then every 28 days). All patients were followed up every 4 weeks for the first 3 months and every 4 to 8 weeks thereafter, or more often if required based on clinical need. Dose reduction and treatment discontinuation were managed as per label indication.

### 2.3. Study Objectives

Primary objective was the incidence of diarrhea related to abemaciclib treatment in our cohort. Secondary objective was the rate of effective management of diarrhea using supportive care only, namely with no need for abemaciclib dose reduction and/or discontinuation. Supportive therapy with loperamide was administered according to guidelines (4 mg of loperamide as a starting dose and then 2 mg as a maintenance dose for a maximum of 16 mg per day, equivalent to 8 capsules). The frequency and severity of diarrhea and the rate of discontinuation and dose reduction were calculated. Exploratory analysis on the impact of gastrointestinal toxicity and its severity on progression-free survival (PFS, defined as the time from initiation of abemaciclib treatment to disease progression or death, whichever occurred first) was conducted.

### 2.4. Statistical Analysis

Data were analyzed using descriptive statistics. Clinical data were summarized as frequencies and percentages or as medians and relative ranges. Mean and standard deviation were used to evaluate the primary objective. Univariate statistical analysis was performed through chi-squared or Fisher’s exact test when appropriate. PFS was evaluated with the Kaplan–Meier method and the differences between groups with the log-rank test (Figure 1). Significance was set at a p-value of 0.05 (2-sided). All analyses were performed using STATA software version 15.

## 3. Results

### 3.1. Patient Characteristics

A total of 43 consecutive female patients with HR+/HER2- advanced breast cancer who received abemaciclib and endocrine therapy were retrospectively reviewed. Of these, four patients were excluded because of missing safety data (two patients were lost to follow-up, one patient moved to another institution and one patient died before starting treatment). In total, 39 patients were included in the analysis (Figure 2). Baseline patient characteristics are reported in Table 1.

### 3.2. Gastrointestinal Toxicity

Table 2 summarizes the data on gastrointestinal toxicity and its management in the patients included in the present analysis. Overall, 36 patients (92%) had diarrhea of any grade and 6 (17%) with a grade ≥3. We observed no life-threatening diarrhea and none of our patients required hospitalization due to diarrhea. Diarrhea was observed together with one (n = 15/36, 42%) or more (n = 15/36, 42%) adverse events in 30 patients (n/36, 83%). The other most common adverse events of any grade were fatigue (33%), neutropenia (33%), emesis (28%), abdominal pain (20%), and hepatotoxicity (13%).

Of the 36 patients who experienced diarrhea, 18 (50%) had G1 diarrhea, 12 (33%) had G2 diarrhea, and 6 (17%) patients reported G3 diarrhea.

Most patients who experienced diarrhea received supportive therapy with loperamide (72%, 26/36).

Figure 3 shows the relationship between the type of supportive treatment and dose reduction or treatment discontinuation in the 36 patients who had diarrhea of any grade.

Loperamide alone was taken by 16 patients because of G1 (n = 6), G2 (n = 7), or G3 (n = 3) diarrhea. Among them, five of them had effective control of diarrhea without dose reduction, two patients had to reduce abemaciclib dose due to diarrhea (the other four patients reduced dose due to emesis G3 and hepatotoxicity G2), and two patients discontinued treatment because of diarrhea (the remaining three patients discontinued abemaciclib because of recurrent neutropenia G3 and hepatotoxicity G3).

Loperamide in combination with other drugs (scopolamine or probiotics) was taken by 10 patients because of G1 (n = 3), G2 (n = 5), or G3 (n = 2) diarrhea. Among them, one patient had effective control of diarrhea without dose reduction, seven patients had to reduce the dose due to diarrhea, none discontinued treatment, while two patients reduced abemaciclib due to other reasons (hepatotoxicity G3 and recurrent neutropenia G3).

The remaining 10 patients who reported diarrhea did not take loperamide because they preferred to manage diarrhea through dietary changes only (8 patients) or medications other than loperamide (2 patients who received diosmectite or probiotics, respectively).

Overall, the success rate of supportive therapy with loperamide (with/without other drugs) was 57.6% (15/26). Of note, this percentage also includes those patients for whom the abemaciclib dose was reduced for reasons other than diarrhea, but all of them had effective management of diarrhea. Loperamide was effective in controlling G1 but not G2/G3 diarrhea (*p* = 0.01). The addition of other drugs to loperamide appeared to have no statistically significant impact on diarrhea control (*p* = 0.12), since in these patients no benefit was observed from the combination. Of note, after dose reduction, all patients reported occasional grade 1 diarrhea, which was easily treated with supportive measures if needed. The median time to treatment discontinuation was 45 days (IQR 14-67). Thereafter, four patients switched to another CDK 4/6 inhibitor, palbociclib, while the other two patients continued treatment with endocrine therapy alone.

Median PFS of the overall population was 12.8 months (range, 1.0–32.4). No differences in PFS were observed for different grading of diarrhea (*p* = 0.83) or for patients experiencing diarrhea versus those not reporting it (*p* = 0.57).

## 4. Discussion

In our retrospective, real-world analysis, we observed a higher incidence rate of diarrhea of any grade and grade 3 compared with data reported in clinical trials.

In a pooled analysis of the MONARCH 1, 2, and 3 studies [11], which included 900 patients, and in a safety analysis of the MONARCH 2 and 3 studies [14], diarrhea of any grade was observed in 82% and 86% of patients, respectively, compared with 92% in our study. In addition, G3 diarrhea occurred in 17% of our patients, which is slightly higher compared with the 12–13% G3 diarrhea reported in the aforementioned studies. Furthermore, permanent treatment discontinuation (10%) and dose reduction due to gastrointestinal toxicity (31%) reported in our study were higher than the data from the clinical trials (3% and 19%, respectively) [14].

Notably, in our cohort, the majority of patients who experienced diarrhea (72%) received supportive therapy with loperamide, alone or in combination with other medications such as scopolamine, diosmectite, and probiotics. Loperamide was significantly active in controlling G1 diarrhea, with an overall success rate of about 58%. Although the recommendations for the management of diarrhea (Figure 4 and Figure 5) suggest that loperamide should be started at the first sign of loose stools (at any grade of diarrhea), in our cohort only 26 out of 36 patients who experienced diarrhea received loperamide. This was mainly due to patient preferences and the fear of most of them that they may suffer from subsequent constipation if the medication was taken prematurely.

One possible explanation for the higher incidence of diarrhea we observed can be found in the differences existing between the patient population enrolled in clinical trials and those treated in the real-world setting. Considering that the population included in clinical trials is generally highly selected and healthier, the safety of new drugs must also be evaluated in real-world patients who might experience more adverse events due to a greater number of concomitant diseases, concurrent medications, lower performance status, and older age and/or other frailty conditions. Interestingly, in the pooled analysis by Modi and colleagues [11], older age (>70 years old) was significantly associated with an increased risk of grade ≥3 diarrhea (HR [95%CI] 1.72 [1.14–2.58], *p* = 0.009). In our analysis, ten patients (26%) were older than 70 years, and all experienced diarrhea (three patients G1, four patients G2, and three patients G3).

In a safety analysis of the MONARCH 2 and 3 trials, Rugo and colleagues showed that the PFS benefit of abemaciclib was not affected by dose reductions or early onset of toxicities [14]. In our cohort, we did not observe differences in PFS among patients with different severity of gastrointestinal toxicity. Of note, all patients who permanently discontinued abemaciclib due to gastrointestinal toxicity were candidates for another CDK4–6i (i.e., palbociclib).

There are already some studies in the literature that have investigated the safety of CDK 4/6 inhibitors in a real-world setting. The study conducted by Carter et al. [17], to evaluate the use of abemaciclib in patients with HR+/HER2- advanced breast cancer within the first year of FDA approval, reported a lower incidence of diarrhea of any grade compared with clinical trials (67% vs. ~85%), and a higher number of treatment discontinuation (12% vs. <3%) and hospitalization due to diarrhea (6.8%). The authors attempted to explain these contrasting results compared to pivotal trials with the differences associated with heterogeneity in the management of adverse events in clinical practice. While physicians must follow strict protocols for dose reductions in clinical trials, this is not happening in the real-world setting, where doctors are more likely to comply with patients’ reported adverse events. Another real-world analysis of a retrospective cohort of metastatic breast cancer patients treated with CDK 4/6 inhibitors [18] found a higher discontinuation rate of 34% in patients treated with abemaciclib. Price et al. [19], analyzed a retrospective cohort of 142 advanced breast cancer patients treated with abemaciclib and reported a significantly higher rate of diarrhea of any grade (43%) that caused a higher discontinuation rate (18%). Finally, Queiroz et al. [20] analyzed a retrospective cohort of advanced breast cancer patients treated with CDK 4/6 inhibitors, 21 of whom were treated with abemaciclib. They reported a higher rate of dose reduction due to adverse events (45%) and 81% of patients experienced diarrhea of any grade with almost half of them (45%) having to reduce the dose.

Compared with the aforementioned studies and other retrospective analyses [17,18,19,20,21,22], which did not distinguish between different grades of adverse events and did not report in detail the type of drugs used to treat abemaciclb-induced diarrhea, our study has several valuable aspects. First, we examined in more detail the classification of each adverse event according to the CTCAE, which is not usually done in daily clinical practice. Second, in our analysis we investigated the other drugs used to treat diarrhea besides loperamide, which are not usually reported in clinical trials but seem to be very common in the real-world population. We also provided very detailed information about the patients’ concomitant diseases that might be related to diarrhea, and we tried to explain the possible causes for the occurrence of this adverse event. Finally, we examined the outcome of patients who received supportive therapy for diarrhea and assessed the success rate of these medications.

The approval of abemaciclib for the adjuvant treatment of high-risk early stage breast cancer on the basis of the MonarchE trial will lead to a gradual increase in the number of patients receiving this drug [10]. A recent safety analysis by Rugo and colleagues on nearly three thousand patients treated with abemaciclib in the adjuvant setting confirmed the safety profile of this therapy but reported a higher level of treatment discontinuation without prior dose reduction, mostly due to low-grade diarrhea [23]. The reasons for early treatment discontinuation are various, ranging from fatigue due to prolonged adjuvant therapies to a lack of knowledge about the real benefits of this strategy and the correct management of potential adverse events. These findings highlight the importance of early and proactive management of diarrhea, especially in a potentially curative setting of patients, by offering them supportive medications and reducing the dose to achieve a tolerable individualized dosage [23].

Diarrhea is one of the side effects that most affect patients’ quality of life and therefore it is essential to manage it correctly.

International guidelines recommend that patients and caregivers should be informed in advance of the possible occurrence of diarrhea and prescribed appropriate supportive therapy. This includes taking two tablets of loperamide at the first sign of loose stools and then one tablet after each evacuation of unformed stools, up to a maximum of eight tablets per day [14,24]. In clinical practice, several mechanisms can be used to prevent the onset of abemaciclib-induced diarrhea. The National Community Oncology Dispensing Association has published a comprehensive guideline for the management of CDK4/6-inhibitor-induced diarrhea, based on real-world experience and scientific evidence [24,25,26]. Key recommendations included maintaining high and adequate fluid intake, eating frequent and small meals, and avoiding lactose, alcohol, and certain irritating foods, such as spicy and fatty foods (Figure 5). Table 3 represents a synoptic table on the guideline-based management of abemaciclib-induced diarrhea.

However, despite taking loperamide, a significant proportion of patients need to reduce the dose, temporarily suspend treatment, or eventually discontinue treatment with abemaciclib due to diarrhea. In addition, the possible side effects associated with the use of antidiarrheal agents, such as constipation, must always be considered. In our analysis, 5% of patients suffered from stypsis after loperamide administration, and this was the main reason why some patients did not follow supportive therapy with loperamide. The addition of other supportive medications such as probiotics to loperamide could potentially be beneficial because of their activities in stabilizing the microbiota. Preclinical studies in murine models have shown that postbiotics may be able to increase fecal IgA levels and stabilize the longitudinal development of microbiota, thereby protecting mucosal surfaces from pathogenic infections and downregulating ongoing inflammatory processes in inflamed tissues [27]. Several pivotal studies are currently underway to investigate the relationship between postbiotics and abemaciclib-induced diarrhea.

Postbiotics have not yet been included in the current guideline recommendations for the management of abemaciclib-induced diarrhea, probably due to a lack of robust data. However, the scenario will rapidly change in the future as some studies are testing the possible utility of this class of agents for the treatment of diarrhea with encouraging results. For example, in a phase II study, Masuda et al. investigated whether the combination of bifidobacterial and trimebutine maleate can reduce the incidence of abemaciclib-induced diarrhea without increasing constipation. Women were randomly assigned to receive either bifidobacterial alone or bifidobacterial and trimebutine maleate, together with abemaciclib and endocrine therapy. The study showed that bifidobacterial with or without trimebutine maleate shortened the duration of abemaciclib-induced diarrhea and prevented the occurrence of grade 3 or higher diarrhea, resulting in a lower rate of drug reduction or interruption. However, no beneficial effect was observed in reducing constipation [28].

The reasons that may explain why abemaciclib causes higher gastrointestinal toxicity compared to other CDK4/6 inhibitors might be related to its pharmacokinetics. More specifically, more than 81% of the drug is fecally excreted and, during this process, active metabolites are produced that play an important role in the occurrence of diarrhea. Unlike the other CDK4/6 inhibitors, abemaciclib also acts on CDK9, an important regulator of intestinal cell proliferation. In preclinical models, several morphological changes of the gastrointestinal tract were observed in rats treated with abemaciclib, such as proliferation of crypt cells, degeneration of enterocytes, and inflammation of the mucosa [29]. However, the pathogenicity of diarrhea is complex and multifactorial, and many factors appear to be involved, such as the activation of the Wnt/β-catenin pathway and upregulation of other solute transporters (Slc28a1, Slc37a2, and Slc5a12), ultimately resulting in cell proliferation [30]. Another important mechanism involved in gastrointestinal toxicity is the inhibition of the glycogen synthase kinase-3 beta (GSk3β), which is a part of a protein complex that phosphorylates β-catenin [31]. Moreover, abemaciclib may inhibit Ca2+/calmodulin-dependent protein kinase CAMKII, a mechanism involved in gut motility that could partially explain the increased bowel motility associated with abemacicilib [29]. Understanding these complex mechanisms may be important for developing safer drugs and strategies to treat diarrhea and thereby improve patients’ quality of life.

Our study has some limitations that should be taken into account. First, it is a retrospective study conducted on a relatively small cohort of patients. Second, the PFS value is lower than those reported in clinical trials and could be due to several real-world factors: nearly two-thirds of patients had extra bone disease and more than half had endocrine resistance disease, two conditions associated with poorer prognosis. In addition, our cohort reflects a real-world population, since about 18% of patients had ECOG PS equal to or greater than 2 and more than one-third of patients were older than 65 years. On the other hand, our work has several strengths: it is a real-world study conducted in a very homogeneous cohort of patients treated with abemaciclib plus endocrine therapy at a single institution, and it has a relatively long follow-up period (~34 months). In addition, we sought to go further than the single description of diarrhea and adverse events in patients treated with abemaciclib. In fact, we tried to establish a possible association between the occurrence of diarrhea and patients’ comorbidity; we aimed to detect the success rate of patients treated with abemaciclib and anti-diarrheal medication; we investigated other drugs that could potentially have a good effect in treating diarrhea with fewer side effects, such as postbiotics; we provided the possible pharmacokinetic explanation for diarrhea associated with abemaciclib.

In general, real-world studies like ours are useful to reassure the medical community about the results of clinical trials and to confirm the safety and efficacy of new medications in a real-world population.

**Figure 4 jcm-12-01775-f004:**
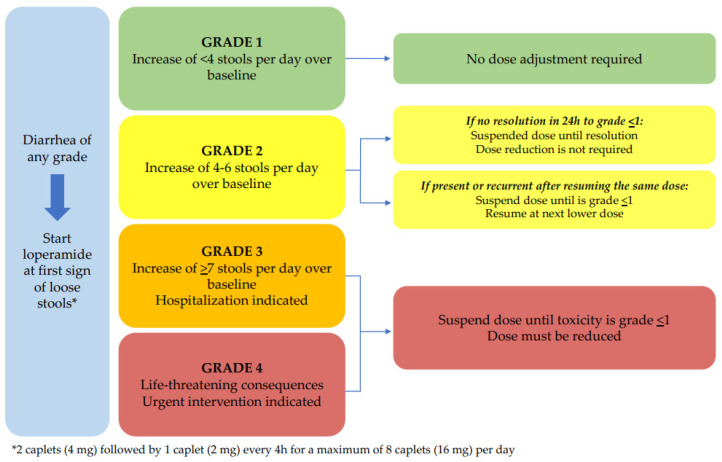
Simplified representation of management recommendations for diarrhea induced by abemaciclib, based on the European Medicines Agency (EMA) recommendations [32]. Toxicity grade is defined according to the Common Terminology Criteria for Adverse Events (CTCAE) v5.0.

**Figure 5 jcm-12-01775-f005:**
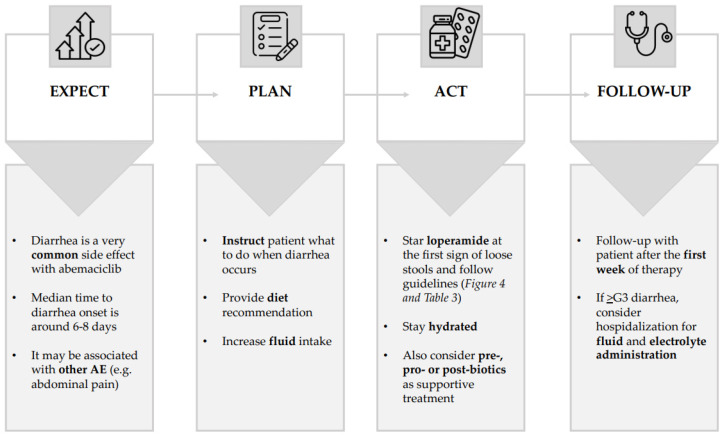
Simplified representation of the four steps in the management of abemaciclib-induced diarrhea.

**Table 3 jcm-12-01775-t003:** Synoptic table on the guideline-based management of abemaciclib-induced diarrhea. Table adapted from summary of product characteristics of abemaciclib of European Medicines Agency (EMA) [32].

Management Recommendations for Diarrhea		
Start loperamide at first sign of loose stools → 2 capsules (4 mg) followed by 1 capsule (2 mg) every 4 h for a maximum of 8 capsules (16 mg) per day		
Grade 1	<4 stools/day	No dose adjustment required
Grade 2	4–6 stools/day	No dose adjustment requiredIf toxicity does not resolve within 24 h to Grade 1 or lower, suspend abemaciclib dose until resolution
Grade 2 that persists or recurs after resuming the same dose despite maximum supportive measures		Suspend abemaciclib dose until toxicity resolves to Grade 1 or lowerResume at next lower dose
Grade 3 or 4	≥7 stools/day or hospedalization	

## 5. Conclusions

In our retrospective, real-world analysis, we observed a higher incidence of diarrhea in patients treated with abemaciclib and endocrine therapy for HR+/HER2- advanced breast cancer. In addition, the rate of permanent treatment discontinuation due to gastrointestinal toxicities was higher than reported in clinical trials. It is of paramount importance to evaluate the incidence of abemaciclib-induced diarrhea in the real clinical scenario, as patients enrolled in clinical trials often do not reflect the real-world population, since they are highly selected patients who may tolerate the treatments better than patients treated daily in the practice setting. Better management of abemaciclib-induced diarrhea is warranted to improve treatment adherence.

## Figures and Tables

**Figure 1 jcm-12-01775-f001:**
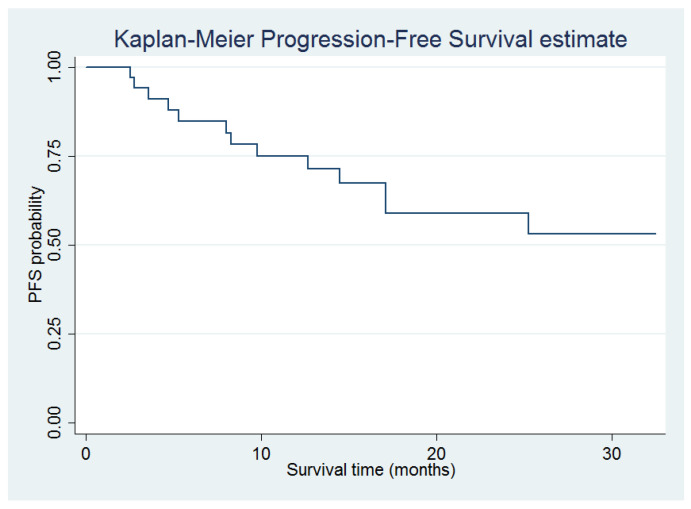
Kaplan–Meier Curve of progression-free survival.

**Figure 2 jcm-12-01775-f002:**
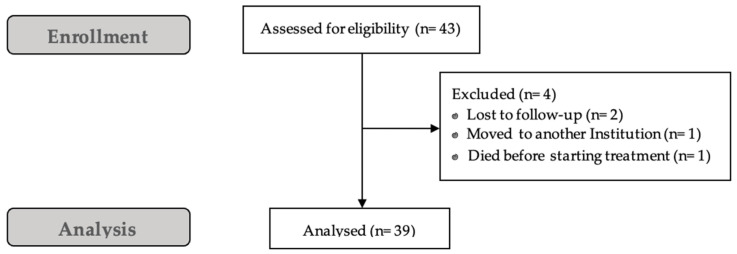
Consort diagram.

**Figure 3 jcm-12-01775-f003:**
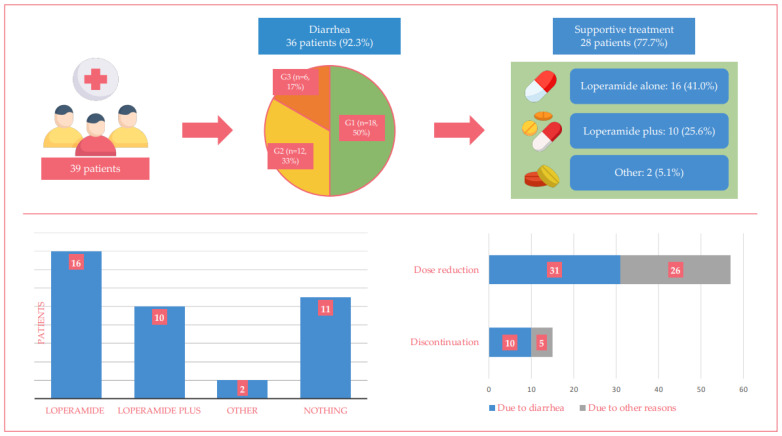
The figure shows the incidence, severity of diarrhea, and type of supportive therapy among the patients enrolled in the study. The histogram at the bottom left shows the type of supportive treatment administered, while the bar chart at the bottom right shows the percentage of dose reduction and treatment discontinuation in the 36 patients who had diarrhea of any grade.

**Table 1 jcm-12-01775-t001:** Baseline patients’ characteristics.

Characteristics			N (%)
All			39 (100)
Age, median (range)			64 (41–82)
Weight, median (range)			62 (38–100)
BMI	≤18		3 (7.6)
	18.5–24.9		15 (38.4)
	25–29.9		7 (17.9)
	≥30		6 (15.3)
	NA		8 (20.5)
ECOG PS	0		21 (53.8)
	1		11 (28.2)
	≥2		7 (17.9)
Smoking history			10 (25.6)
Comorbidities	No		18 (46.1)
	Yes	Total	21 (53.8)
		1	8 (20.5)
		≥2	13 (33.4)
	Cardiovascular:		
	Hypertension		11 (52.3)
	Deep vein thrombosis		3 (14.2)
	Ischemic heart disease		1 (4.7)
	Arrhythmias		1 (4.7)
	Metabolic:		
	Diabetes		2 (9.5)
	Obesity		6 (28.5)
	Dysthyroidism		2 (9.5)
	Dyslipidemia		4 (19.0)
	Neurological:		
	Stroke/TIA		3 (14.2)
	Schizophrenia		1 (4.7)
	Depression		1 (4.7)
	Paraparesis		1 (4.7)
	Intestinal:		
	Inflammatory bowel disease		1 (4.7)
	Diverticulosis		1 (4.7)
	Celiac disease		1 (4.7)
	Pulmonary:		
	Asthma		1 (4.7)
	Reumatological:		
	Arthrosis		2 (9.5)
	Gynecological:		
	Endometriosis		1 (4.7)
	Uterine fibroma		1 (4.7)
Metastatic site	Visceral disease	Total	21 (53.8)
		Liver involvement	12 (30.7)
	Bone only disease		13 (33.3)
	Lymphonodal disease		5 (12.8)
Prior adjuvant chemotherapy			15 (38.4)
Line of therapy	1st		37 (94.8)
	2nd or more		2 (5.2)
Endocrine therapy	Letrozole		19 (48.7)
	Fulvestrant		20 (51.2)
Endocrine resistance	No		18 (46.1)
	Primary		10 (25.7)
	Secondary		11 (28.2)

Abbreviations: NA: not available, ECOG PS: Eastern Cooperative Oncology Group performance status, BMI: body mass index, PS: performance status, TIA: transient ischemic attack.

**Table 2 jcm-12-01775-t002:** Abemaciclib-induced toxicities and management.

Toxicities and Management		N (%)
All patients		39 (100)
Any AEs	No AE	1 (2.5)
	Only diarrhea	6 (15.5)
	Aes without diarrhea	2 (5.0)
	Diarrhea and 1 AE of any grade	15 (38.5)
	Diarrhea and ≥2 AE of any grade	15 (38.5)
Diarrhea	Total	36 (92.3)
	Grade 1	18 (50.0)
	Grade 2	12 (33.3)
	Grade 3	6 (16.7)
Type of other Aes of any grade	Fatigue	13 (33.3)
	Neutropenia	13 (33.3)
	Emesis	11 (28.2)
	Abdominal pain	8 (20.5)
	Hepatotoxicity	5 (12.8)
	Hyporexia	4 (10.2)
	VTE	2 (5.1)
	Constipation	2 (5.1)
Supportive therapy		28 (77.7)
Type of supportive therapy	Loperamide alone	16 (41.0)
	Loperamide with other medications *	10 (25.6)
	Other medications * without loperamide	2 (5.1)
	No supportive therapy	11 (28.2)
Success rate of loperamide alone or with/without other drugs		15/26 (57.6)
Abemaciclib dose reduction	Overall	22 (56.4)
	Due to diarrhea	12 (30.8)
	Due to neutropenia	4 (10.3)
	Due to PE	2 (5.1)
	Due to hepatotxicity	2 (5.1)
	Due to astenia	2 (5.1)
Treatment discontinuation	Overall	6 (15.3)
	Due to diarrhea	4 (10.2)
	Due to hepatotoxicity	1 (2.5)
	Due to emesis	1 (2.5)
Median time of treatment discontinuation		45 days (IQR 14–67)
Treatment after discontinuation	Switch to palbociclib	4 (10.2)
	Continue only ET	2 (5.1)

Abbreviations: AEs: adverse events, ET: endocrine therapy, PE: pulmonary embolism, VTE: venous thromboembolic events. * other medications were scopolamine (21.4%), probiotics (17.8%), and diosmectite (7.1%).

## Data Availability

Restrictions apply to the availability of these data. The data presented in this study are available on reasonable request from the corresponding author.

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
