# Peer review of "Sticking to the Rules: Outcome and Success Rate of Guideline-Based Diarrhea Management in Metastatic Breast Cancer Patients Treated with Abemaciclib"

_jcm, 2023, doi:10.3390/jcm12051775_

Round 1
Reviewer 1 Report
In this study, the author aimed to determine whether the incidence of abemaciclib-induced diarrhea in real-world was higher than the one reported in clinical trials, where patients are highly selected, and to evaluate the success rate of standard supportive care in this setting. However, this study looks quite similar with many other recent studies [1-2].
[1]. Masuda N, Chen Y, Kawaguchi T, Dozono K, Toi M. Safety in Japanese Advanced Breast Cancer Patients Who Received Abemaciclib in MONARCH 2 and MONARCH 3: Assessment of Treatment-Emergent Neutropenia, Diarrhea, and Increased Alanine Aminotransferase and Aspartate Aminotransferase Levels. Cancer Manag Res. 2022;14:1179-1194.
[2]. Sammons S, Moore H, Cushman J, Hamilton E. Efficacy, safety and toxicity management of adjuvant abemaciclib in early stage HR+/HER2- high-risk breast cancer. Expert Rev Anticancer Ther. 2022 Aug;22(8):805-814. doi: 10.1080/14737140.2022.2093719. Epub 2022 Jun 30. PMID: 35737886.
Author Response
Thank you for your feedback. We know that there are already other examples in literature of retrospective studies on the management of abemaciclib-related adverse events. However, we believe that our analysis provides some interesting aspects for the physician’s community, especially considering that abemaciclib is becoming more common in clinical practice as it has also been approved for high-risk HR+ early breast cancer and thus a larger number of patients are treated with this drug.
However, we appreciate your feedback and have cited the two retrospective analyses you suggested, to highlight that this is a very actual topic and that results may be contradictory. Indeed, the incidence of abemaciclib-induced diarrhea is very much dependent on clinical and medical management, which can vary widely from hospital to hospital and country to country.
We have also tried to highlight the strengths of our work compared to previously published studies, which are the following:
- it is a cohort of consecutive and homogeneous breast cancer patients
- we provided detailed information on patients’ comorbidities that might be related to diarrhea
- we tried to explain possible causes for the occurrence of diarrhea
- we tried to provide useful indications on how to treat this adverse event
- we investigated the drugs used to treat diarrhea, apart from loperamide
(line 331-338)

Reviewer 2 Report
This study describes a retrospective cohort observational study of abemaciclib in combination with endocrine therapy in 39 patients with metastatic breast cancer. The authors state that guideline-based management of adverse events of abemaciclib therapy, particularly diarrhea and gastrointestinal toxicity, is important in clinical practice. My comments are as follows.
1. This study is a retrospective, real-world analysis of abemaciclib and endocrine therapy for metastatic breast cancer. The management of adverse events with abemaciclib follows guidelines and is more detailed than in previous clinical trials in which patients were highly selected. However, no scientific priorities were found.
2. PFS is important in evaluating the efficacy of abemaciclib and endocrine therapy for metastatic breast cancer; a survival curve for PFS should be illustrated.
3. In Table 1, Lynfonodal disease is incorrect; Lymphonodal disease is correct.
Author Response
Thank you for your comment. We are aware that similar retrospective analyses on real-world patients treated with abemaciclib have already been published in literature in recent years. However, we believe that our work may be relevant to the oncology community for several reasons. First, abemaciclib will become more widely used in clinics as it has been approved for the treatment of early-stage, high-risk breast cancer, and thus more women may face the potential side effects of this drug. Second, we acknowledge that our analysis has some strengths compared to other studies:
- this is a homogeneous cohort of consecutive breast cancer patients
- we have captured the patients' concomitant diseases that might be related to diarrhea in some way
- we provide information on the possible mechanism of action related to the occurrence of diarrhea
- we investigated other medications used to treat diarrhea, besides loperamide, with particular interest in postbiotics
(line 331-338)
Thank you for your comment. We agree with you that assessing PFS in patients with metastatic breast cancer is extremely relevant, especially when facing moderate/severe toxicities. Indeed, we have included the Kaplan-Meier curve for PFS as suggested (please see Figure 3).
We have corrected “Lynfonodal” with “Lymphonodal” disease.

Reviewer 3 Report
Dear authors,
thank you for this valuable analysis to a clinical relevant topic. As diarrhea is a common adverse effect under abemaciclib and the prescription rates of abemaciclib are increasing it is important for the clinicians to become information concerning the real world data in this topic.
I appriciate that you provided detailed information on the following topics:
- possible mechanisms of the development of diarrhea under abemaciclib
- comorbidities of the included patients
- a detailed report concerning further applied ( beyond loperamide) treatments for diarrhea
- The detailed information in the table 2
- the explanation of the higher rate of diarrhea in your evaluation comparing to the clinical study for drug approvement
This points show that your clinical evaluation was carefully planed.
I would like to suggest you to add following information:
- a short summary of the guidelines information concerning the diarrhea management, beyond loperamid
- and please provide the information if a treatment in the hospital was needed in your trial
Kind reagards,
Author Response
Reply: We appreciate that you value our study and have recognised the key strengths of our analysis. We have accepted your suggestion and added to the discussion more insight into the current guidelines on diarrhea management beyond loperamide, in particular the potential beneficial effects of postbiotics. To this end, we have cited the results of MERMAID trial, which tested the association of bifidobacterial and trimebutine maleate (line 291-303)
As described in table 2, none of our patients experienced life-threatening diarrhea. However, we recognized that this might be an interesting point to comment in our study and so we added it to the results (line 157-158).

Round 2
Reviewer 1 Report
Thanks for the feedback. As the author mentioned, this paper has some strengths compared to some previously published papers. They explored different treatments and combinations for diarrhea and tried to explore its causes. Even though this paper is quite similar to some previously published papers, it had some strengths indeed. My suggestions are as follows:
1. It would be better if the author could make a summary table of guidelines about the treatment of diarrhea management in abemaciclib.
2. Add more discussion about other clinical trials that are associated with diarrhea management with abemaciclib. And discuss the strengths and advantages of your study compared with other studies.
Author Response
1) We appreciate your insightful feedback. We have added a synoptic table on the guideline-based management of abemaciclib-induced diarrhea (Table 3, line 427). Furthermore, we tried to provide an easy-to-read figure for the recommended approach to this very common adverse event (Figure 4, line 422).
2) Thank you for your suggestions. We have reviewed the main clinical trials and retrospective real-world studies on the safety of abemaciclib. Then, we have reported the most interesting findings from real-world data and compared them with clinical trials. Furthermore, we highlighted the strengths of our work compared to the abovementioned studies (please see lines 245 to 276) and discussed them at the end of the discussion (please see lines 374 to 384).
